# Pivotal Role of Inflammation in Celiac Disease

**DOI:** 10.3390/ijms23137177

**Published:** 2022-06-28

**Authors:** Maria Vittoria Barone, Renata Auricchio, Merlin Nanayakkara, Luigi Greco, Riccardo Troncone, Salvatore Auricchio

**Affiliations:** 1Department of Translational Medical Science, University Federico II, Via S. Pansini 5, 80131 Naples, Italy; r.auricchio@unina.it (R.A.); merlin.nanayakka@unina.it (M.N.); ydongre@unina.it (L.G.); troncone@unina.it (R.T.); 2European Laboratory for the Investigation of Food Induced Disease (ELFID), University Federico II, Via S. Pansini 5, 80131 Naples, Italy; salauric@unina.it

**Keywords:** Mediterranean and Western diet, celiac disease, gluten, inflammation, microbiota, viral infections

## Abstract

Celiac disease (CD) is an immune-mediated enteropathy triggered in genetically susceptible individuals by gluten-containing cereals. A central role in the pathogenesis of CD is played by the HLA-restricted gliadin-specific intestinal T cell response generated in a pro-inflammatory environment. The mechanisms that generate this pro-inflammatory environment in CD is now starting to be addressed. In vitro study on CD cells and organoids, shows that constant low-grade inflammation is present also in the absence of gluten. In vivo studies on a population at risk, show before the onset of the disease and before the introduction of gluten in the diet, cellular and metabolic alterations in the absence of a T cell-mediated response. Gluten exacerbates these constitutive alterations in vitro and in vivo. Inflammation, may have a main role in CD, adding this disease tout court to the big family of chronic inflammatory diseases. Nutrients can have pro-inflammatory or anti-inflammatory effects, also mediated by intestinal microbiota. The intestine function as a crossroad for the control of inflammation both locally and at distance. The aim of this review is to discuss the recent literature on the main role of inflammation in the natural history of CD, supported by cellular fragility with increased sensitivity to gluten and other pro-inflammatory agents.

## 1. Introduction

Chronic inflammatory diseases are increasing and becoming a social and medical issue. The big family of increasing chronic inflammatory diseases (obesity, diabetes, inflammatory bowel diseases, autoimmune diseases, cancer and several ageing-linked diseases) is generated by the meeting of several different factors, such as environmental, genetic and microbiota [1]. Among the environmental factors that induce inflammation, a main role is played by nutrients, which by themselves, can have pro-inflammatory or anti-inflammatory effects, directly or indirectly mediated by the intestinal microbiota [2]. Dietary carbohydrates [3], lipids [4], salt [5], proteins [2] and in general the “Western lifestyle” [6] can induce inflammation in cells. Rural Africans consume substantially more fiber than people in Western countries and rarely develop diseases such as allergies, asthma, colon cancer, or cardiovascular diseases [7]. The Western diet comprises refined grains, alcohol, salt, certain oils, corn-derived fructose, fatty domesticated meats, and energy dense and processed foods frequently consumed in caloric excess. It can be further characterized by the reduced consumption of fruits and vegetables. In general, the Western diet and lifestyle have been long linked to low-grade inflammation that represents the “common background” of several different diseases [8].

The intestines appear to be a cross-road for the control of inflammation, both locally and at distance [9]. Both in obesity an insulin resistance, the intestinal immune system plays a pivotal role [10]. Obesity and insulin resistance predisposes one to altered intestinal immunity and is associated with changes to the gut microbiota, intestinal barrier function, gut-residing innate and adaptive immune cells, and oral tolerance to luminal antigens. Accordingly, the gut immune system may represent a novel therapeutic target for systemic inflammation in insulin resistance. Moreover, IgA antibodies mobilized from the gut play an unexpected role in suppressing neuroinflammation, demonstrating that intestinal inflammation can influence many organs at close range and at distance [9].

A new concept is arising in the recent literature in which a fragile cell can be predisposed to autoimmune diseases. Genetic predisposition can render the cells more prone to inflammatory triggers, creating cellular vulnerability. A notable example of this mechanism is beta cell fragility, which has been demonstrated to underlay both type 1 and type 2 diabetes [11].

Celiac disease (CD) is a good example of an inflammatory chronic disease induced by food. It is an immune-mediated enteropathy triggered in genetically susceptible individuals by a group of wheat proteins and related prolamins from cereals [12]. It is characterized by a variable combination of gluten-induced symptoms, generation of CD-associated autoantibodies, and enteropathy [13]. A central role in the pathogenesis of CD is played by the HLA (human leukocyte antigen)-restricted gliadin-specific intestinal T cell response [12,14]. Although activation of T cells has been studied in depth, the central question still remains unanswered, namely, why a pro-inflammatory T cell response is generated toward gliadin instead of a regulatory response, which normally promotes oral tolerance to dietary protein antigens. When, how and where this inflammation is generated in CD is not clear. The answer partially comes from studies conducted in mice that demonstrated that mucosal inflammation due to reovirus infection may disrupt oral tolerance by suppressing regulatory T cell conversion and by promoting Th1 immunity against gliadin [15], indicating that in an inflamed environment enriched in cytokines, such as IL-15 or type I interferons, T cells tend to acquire a pro-inflammatory phenotype. Many mice-based studies as well as epidemiological data have suggested viral infections as one of such factors [16]. This review aims at discussing: (1) the available evidence of constitutive fragility of the intestines in CD; (2) the susceptibility of celiac patients and cells to gliadin and other pro-inflammatory triggers; and (3) future prospective.

## 2. Results

### 2.1. Celiac Disease as an Inflammatory Chronic Disease

The recent literature describes in CD a meeting of several different factors such as cellular vulnerability, pro-inflammatory effects of gluten and other wheat proteins, Western diet, and other environmental triggers such as viruses that prepare and/or amplify the T cell-mediated response to gluten. The factors that create a pro-inflammatory environment in the CD intestines, leading to an expansion of gliadin-specific T cells in genetically susceptible individuals and further shifting them toward a pro-inflammatory phenotype, could have multiple origins: the pro-inflammatory environment (exogenous stimuli), such as diet, viruses and other pro-inflammatory factors; and the constitutive cellular alterations (endogenous predisposition) that by themselves induce and/or render the cells more sensitive to pro-inflammatory stimuli. All these factors, both exogenous and endogenous, can contribute to the generation of “sterile” inflammation in CD (Figure 1). In the following paragraphs, we analyze the endogenous alterations that could create a constitutive inflammatory predisposition, and among the exogenous factors, we focus mainly on nutrients: in general, wheat proteins, and in particular, viruses.

### 2.2. Endogenous Alterations in CD Independent of Gluten

Endogenous alterations, defining a celiac cellular phenotype, have been described in different tissues and cells from CD patients, including intestinal organoids [17]. Gluten exacerbates these constitutive alterations by increasing the same markers already altered in the absence of gluten, both in vitro and in vivo. This phenotype confers the vulnerability to the CD cells to several different triggers that have effects on different pathways, including innate immunity activation. Many of these constitutive alterations are now regarded as biomarkers of clinical relevance, as they can be used to intervene in the “at risk” population before the onset of the disease [17,18].

The recent literature is starting to address the question of endogenous alterations, independent of gluten, in CD by in vivo studies and in cellular models. From these studies, it clearly appears that subtle alterations of the CD cells are also present in the absence of gluten as well as in the absence of T cell activation.

We describe the literature concerning these points by starting with population studies, such as patients at risk of CD, CD patients with GFD–CD (gluten-free diet–celiac disease) and in vitro studies on biopsies, intestinal organoids and CD cells from compartments different from the intestines (Table 1).

#### 2.2.1. Patients at Risk of CD: What Happens before the Disease?

Patients at risk of CD are those subjects that have a genetic risk defined as the presence of HLA DQ2 or 8 and are first-degree relatives (sons or daughters, sisters or brothers) of a CD patient. These subjects were followed from birth in prospective studies for several years. Studies on this population also had the aim to find predictive markers of the disease years before its onset [53] (Table 1).

In a recent study [19], infants at risk for CD were followed up to 8 years to monitor the onset of CD. In infants that developed CD, the serum phospholipid profile was altered at 4 months, long before any exposure to gliadin or any production of anti-tTg (tissue transglutaminase) antibodies, compared to their controls matched for the same genetic profile that during the 8 years of observation did not become celiac patients. The phospholipid signature was dramatically different in infants who developed CD when compared to that of infants at risk not yet developing CD. A specific phospholipid profile was able to discriminate infants who eventually developed celiac disease years before antibodies or clinical symptoms appeared. This lipid profile appears to be constitutive since it is constant over time and is already present at 4 months of age [19]. Other groups have obtained similar results in CD [20]. Most of the lipids found altered in this study perform an important structural role in the plasma membrane, contributing not only to membrane integrity but also to intracellular vesicle functions. Recent studies have shown that vesicle trafficking might be a crucial key in understanding the handling of gliadin peptides in celiac patients [39,54,55] (Table 1). Moreover, lipids not only have structural functions in the composition of membranes and organelles, they also have crucial signaling roles over and above their nutritional function. Therefore, lipid metabolism and its unbalance can contribute to acute and chronic inflammation in many different conditions and ways [56]. Previous lipid omics studies found that dysregulated lipid metabolism precedes islet autoimmunity and overt disease in children who later progressed to T1D, indicating that alterations in lipid metabolism can have a central role in autoimmune diseases [57,58].

The impaired growth of children, as assessed at the time of CD diagnosis, is a common manifestation of CD, although in CD patients on a gluten-free diet, there is no growth impairment. In the at-risk population, alteration in growth has been described before gluten introduction or before the appearance of any signs of the disease (specific antibodies or symptoms). The growth rate of at-risk subjects that developed CD and the controls were normal at birth, but early on, at 4 months, differences were present between the two groups [21].

In these same at-risk subjects, there was low-grade inflammation with an increase in pro-inflammatory cytokines in the serum before gluten introduction. At 4 months before gluten introduction, the levels of a certain set of proinflammatory cytokines were higher in the serum of predisposed infants who developed celiac disease compared to controls. These results suggest the presence of an “inflamed” background before the exposition to gluten. The genetic profile of infants who develop CD shows a number of differential SNPs (single-nucleotide polymorphisms) in the gene associated with CD compared to those who do not develop the disease, up to 6 years [22,23] (Table 1).

The possible hypothesis to explain these early alterations in CD patients is that a genetic predisposition could modulate the serum cytokine pattern before gluten introduction, which might create a dangerous background for the introduction of gluten during the following years.

Finally, a recent study suggests shifts in the early trajectory of the gut microbiota along with changes in immune markers in infants before weaning who later develop CD, indicating deviations of the normal microbiota maturation process before CD development [24] (Table 1).

In conclusion, there are several important implications. In CD, some constitutive alterations in metabolic, inflammatory and structural pathways are present before the onset of the disease, before gluten introduction in the diet, and consequently, before the gluten-induced autoimmune response. Thus, constitutive alterations may contribute to creating a chronic inflammatory condition that precedes and prepares the T cell-mediated response.

#### 2.2.2. GFD–CD Patient Biopsies, Lessons from In Vivo and In Vitro Studies

Celiac patients after diagnosis have to initiate a strict diet without gluten. These patients are defined as gluten-free diet–celiac disease patients (GFD–CD). The dietary restrictions have to be followed for life. GFD–CD patients do not eat gluten; thus, the main cause of the disease is not present, but some alterations mainly at the intestinal level can still be found. In this paragraph, we describe the studies performed in GFD–CD patients and their biopsies.

##### GFD–CD Patients before and after Gluten Challenge In Vivo

New observations have been obtained and published challenging gluten in vivo in GFD–CD patients by analyzing before and after gluten introduction on several different pathways such as proliferation, inflammation and differentiation at the intestinal level [25,26,27].

Dotsenko et al., demonstrated that the intestinal morphology in patients on a strict GFD was similar to that measured in control subjects [25]. However, gene transcription showed that the GFD–CD group differed significantly from the control group, showing a substantial number of differentially expressed genes. Gene Ontology and Reactome pathway analyses revealed that patients on a GFD presented altered expression of genes with functions such as brush border assembly, developmental processes, transport of small molecules, and FOXO (Forkhead box-O)-mediated transcription of cell cycle genes [25].

Moreover, the Wnt (Wingless and Int 1) pathway has been found altered in GFD–CD patients with respect to controls, suggesting that the alterations of a pathway nodal to intestinal differentiation and homeostasis is also present in the absence of gluten [25]. In the gastrointestinal tract, Wnt signaling activation drives homeostasis and damage-induced repair. Wnt signaling also has a key role during stem cell-driven intestinal homeostasis, regeneration, ageing and cancer [59]. After being gluten challenged, the Wnt pathway was further altered, as tested by RNA sec, in intestinal biopsies of patients on a GFD with respect to controls [25]. All together, these data show that even on a strict GFD, patients reveal patterns of ongoing disease and that gluten acts on the same pathways already altered.

Another recent paper by Stamnaes J. et al. [26,27] shows by proteomics analysis that before being challenged in GFD–CD patients, epithelial inflammation is already in the intestines, with changes indicative of minor crypt hyperplasia and low-grade inflammation in the serum. After challenge with gluten, an increase in the same proteins, already altered at base line in GFD–CD responders, was found. Despite clinical and histological remission, celiac disease patients that develop a mucosal response after 14 days of gluten challenge have already at baseline altered protein compositions of their gut tissue with signs of ongoing inflammation.

In conclusion, these observations indicate that in GFD–CD patients, several different pathways such as proliferation, inflammation and differentiation are altered at the intestinal level and that they can be further altered by gliadin challenge.

##### GFD–CD Patient Biopsies

Several papers have been published studying the biopsies of GFD–CD patients. In these papers, several pathways have been found altered by different techniques. Most of the papers analyze the whole biopsies, as in the case of genetic or expression studies, while in some others, only the epithelium are evaluated. The following listed are the most significant literature contributions that we discuss, clustered by the pathways found altered.

A.Inflammation

Several different genetic [28] and expression [29,30,31] studies indicate that the NFkB (nuclear factor kappa-light-chain-enhancer of activated B cells) pathway is altered in CD. It is widely accepted that NFkB is a key regulator of inducible gene expression in the immune system. It has also been shown that NFkB is a mediator of IL15, a major player of innate immunity in CD. Both innate and adaptive immune responses, as well as the development and maintenance of the cells and organs as well as homeostasis that comprise the immune system are, at multiple stages, under the control of the NFkB family of transcription factors. Moreover, NFkB is responsible for the transcription of genes encoding a number of pro-inflammatory cytokines and chemokines [60]. Fernandez-Jimenez et al. [29] showed that the expression of 93 NFkB genes measured by RT-PCR was altered in a set of uncultured active and treated CD patients with respect to control biopsies, and in cultured biopsies from CD at different stages of the disease (GCD–CD and GFD–CD) challenged with gliadin. These results showed that genes constitutively upregulated in GFD–CD patients belonged to the essential core of the pathway and had crucial, regulatory and central roles in the NFkB signaling system, whereas genes that were overexpressed only in active CD appeared to be more peripheral and included mostly NFkB-inducible interleukins, adhesion molecules and receptors. Moreover, gluten challenge on GFD–CD biopsies increased the NFkB pathway [29,30,31]. The lncRNA13 (long non coding RNA 13) levels are significantly decreased in biopsy samples from the small intestines of CD patients, leading to poor control of the repressed pro-inflammatory genes, and possibly contributing to underlying chronic inflammation [30,31].

B.Innate immunity pathways

Stress signals are expressed by the enterocytes of patients with celiac disease [32]. In the duodenal mucosa of children with untreated CD as well as in children with treated CD, compared with those in controls, elevated HSP72 (Heat shock 70 kDa protein 1) mRNA expression and higher protein levels were found [33]. Moreover, innate immunity has been found activated in GFD–CD patients. The most studied cytokines in GFD–CD biopsies are: IL15 (interleukin 15)/IL15R (IL15 receptor) and INF-α (Interferon α) [34,35,36]. IL15 is a cytokine whose role is essential to the control of immune homeostasis; its expression is tightly regulated at the translational, transcriptional, and intracellular trafficking levels [37]. A high number of CD patients on a GFD–CD maintain high levels of IL-15 expression in the epithelium [34]. IL15R alpha mRNA expression and proteins were increased in CD patients, as compared with non-CD controls. Moreover, CD patients show increased sensitivity to IL-15 stimuli to produce both nitrites and IFN (Interferon) gamma [35]. A mouse model bearing the predisposing HLA-DQ8 molecule and that reproduces the overexpression of IL-15 both in the gut epithelium and the lamina propria (LP) features characteristics of active CD, having clarified the role of IL 15 in the pathogenesis of CD. These mice upon gluten load develop villous atrophy (VA) [38]. The link between infections and CD has been established clinically by several large-scale, population-based cohort studies [61]. Another player of the innate immunity response to gliadin in CD seems to be IFN-α and its downstream effector MxA (*Myxovirus* resistance protein A) [34], both involved in the Toll-like receptor response to external agents, such as viral infections. Higher levels of expression of MxA and IFN-α were found in GFD–CD biopsies with respect to controls before and after treatment with gliadin in GFD–CD biopsies [36].

C.Enterocyte proliferation and differentiation

Damage to the intestinal mucosa in CD is mediated both by inflammation due to the adaptive and innate immune response to gliadin and by proliferation of crypt enterocytes as an early alteration of CD mucosa causing crypt hyperplasia [62,63,64]. The celiac intestines are characterized by an inversion of the differentiation/proliferation program of the tissue, with a reduction in the differentiated compartment, up to complete villous atrophy, and an increase in the proliferative compartment with crypt hyperplasia [40,41].

In CD biopsies both from GCD– and GFD–CD patients, increased activity of the EGFR (epidermal growth factor receptor)/EGF (epidermal growth factor) system and the downstream signaling molecule ERK (extracellular signal-regulated kinases) has been described [39,40,42] together with the enhancement of crypt enterocyte proliferation dependent on EGFR/ERK activation [39,40,42].

Gluten and gliadin peptide p31-43 can increase proliferation in GFD–CD patient biopsies both in vitro [54,65] and after gluten challenge in vivo [66].

D.Structural alterations

The intestinal epithelial tight junction (TJ) barrier controls the permeation of the intestinal lumen contents into the intestinal tissue and consequently into systemic circulation. A defective intestinal TJ barrier has been implicated as an important pathogenic factor in inflammatory diseases of the gut, including Crohn’s disease, ulcerative colitis, necrotizing enterocolitis, and celiac disease.

CD-altered intestinal permeability may facilitate the entry of gluten or its fractions into the lamina propria where it can cause a series of immunological events. Permeability of the intestinal mucosa has been tested by several different methods in vivo and in vitro [43,44,45,46,67,68] in CD patients on a GFD. Persistently increased mucosal permeability to certain probes (molecular weight less than 1500 Daltons) has been shown in patients with celiac disease on a GFD with normal intestinal histology [44]. Patients with both celiac disease and dermatitis herpetiformis show persistent increase in intestinal permeability, which suggests a common pathogenetic mechanism for both disorders [43]. Moreover, barrier dysfunction and structural alterations found in acute CD were only partially recovered in GFD–CD patients, suggesting a level of “minimal damage” [45,46] in CD patients on a GFD. Finally, the impact of altered intestinal permeability may have a role in the secondary autoimmune phenomena as well as the extraintestinal manifestations of CD. CD patient’s antibodies can have biological effects on intestinal and other cells. CD patients may present autoantibodies directed against extraintestinal antigens such as antineuronal and antiganglioside antibodies (more prevalent in patients with neurological disorders) [67,68,69]. CD patients on a gluten-containing diet and severe mucosal damage (villous atrophy) frequently present anti-actin IgA antibodies that display high specificity for CD and disappear after GFD, as previously reported [70]. In addition, a link between atopy and CD has been suggested by an Italian study [71].

Alterations at the genetic and expression levels have been found for proteins involved in the regulation of permeability and tight junctions in CD [47,72,73], suggesting that at least partially, this alteration of permeability can be attributed to genetic defects.

A possible alternative or additive mechanism to genetic defects of TJ-altered functionality can be attributed to pro-inflammatory cytokines, which are produced during intestinal inflammation, including interleukin-1β (IL-1β), tumor necrosis factor-α, and interferon-γ, which all have important intestinal TJ barrier-modulating activity [74].

Another structural alteration that has been described in CD enterocytes of GFD–CD patients is the alteration of vesicular trafficking. CD enterocytes from GFD–CD patients present a constitutive alteration in the intracellular vesicular system at the early recycling compartment level. In CD enterocytes, the number of early vesicles is increased, and EGF/EGFR trafficking is delayed in early endocytic vesicles. Moreover, the decay of EGFR is prolonged, and EEA1 (early endosome Antigen1) and TfR (transferrin receptor) levels are increased [39]. Recently, several human diseases characterized by inflammation and/or autoimmunity have been attributed to alterations in vesicular trafficking at various levels [75,76]. In particular, hereditary autoimmune-mediated lung disease in humans was found to be due to alterations in vesicular trafficking, revealing a role for intracellular transport in the induction of human autoimmunity [77].

An analysis of non-HLA loci and/or published eQTL (expression quantitative trait loci) effects associated with CD revealed that 8 out of 127 that had previously been shown to be related to the immune response or other functions were also related to vesicular trafficking [39]. The results of the analysis of the non-HLA loci and/or published eQTL effects support a role for vesicular trafficking in the pathogenesis of CD.

To increase complexity, epigenetic changes such as DNA methylation may also contribute to the mechanisms that lead to the disease or, once the disease has been initiated, establish a chronic disorder. As an example, evaluation of the methylome of the epithelial and immune cell populations of duodenal biopsies in CD patients and healthy individuals showed that these cell populations present different methylation signatures, which may have a broad effect on structural cellular alterations in a cell type-specific manner [23,78]. In CD, the genes linked to inflammatory processes are upregulated, as said before, whereas the genes involved in cell adhesion/integrity of the intestinal barrier are downregulated. All together, expression and methylation studies have highlighted a “gene-expression phenotype” of CD and have shown that the abnormal response to dietary antigens in CD might be related not to abnormalities of gene structure but to the regulation of molecular pathways [23].

In conclusion, studies on the GFD–CD population have helped to give an important insight into the pathogenesis of CD, but their meaning is still controversial. Some researchers regard the alterations found in GFD–CD patients as an effect of residual amounts of gluten in the diet [26]. Although small amounts of gluten contaminant cannot be ruled out from food and the CD intestines may be sensitive even to small amounts of gliadin, new insights from compartments far away from the intestines and intestinal organoids point to the presence of constitutive alterations in CD, as discussed in the next paragraphs.

#### 2.2.3. Intestinal Organoids

An emerging role in CD pathogenesis has been attributed to the intestinal epithelium. In epithelial cells of CD patients, morphological and functional alterations have been described together with the activation of the inflammasome pathway [39,79].

Organoids derived from the small intestine represent a new tool to study the role of the intestinal epithelium in several different diseases [80]. Intestinal organoids are derived from crypt stem cells cultivated in 3D and embedded in a matrix; they resemble the small intestinal epithelium. Organoids from CD patients have shown the presence of increased staminality, permeability, inflammasome activity, and innate immunity genes with respect to organoids in healthy individuals [48,79]. Instead, extracellular matrix (ECM) genes were decreased in [49]. Moreover, increased markers of inflammation were found at the protein and mRNA levels in CD organoids. This inflammation was not a residual effect of the tissue of origin but is probably constitutive, as it was persistent even after many days in culture [17].

In CD biopsies and in intestinal organoids, increased sensitivity to inflammatory stimuli from bacteria [79], viral ligand loxoribine [17] and gliadin peptide P31-43 [17] have been described, indicating that intestinal organoids from CD patients are more sensitive to pro-inflammatory stimuli. Taken all together, these observations indicate that CD intestinal epithelial cells are constitutively different from those in healthy individuals.

#### 2.2.4. Non-Intestinal Cells

In this paragraph, we discuss the literature on CD cells derived from regions different from the intestines in GCD and GFD–CD patients. The read outs that have been investigated in these regions are mostly overlapping with those investigated in the intestinal biopsies. In particular, alterations of inflammation and innate immunity pathways, proliferation and structural alterations, including vesicular trafficking, have been described. Both skin and intestinal fibroblasts derived from CD patients at different stages of the disease have been used to study constitutive alterations in CD. Skin fibroblasts, far away from the main site of the disease, the intestines, are particularly interesting. Inflammation and innate immunity markers such as pNFkB, Mxa, IL15R-alpha, and pEGFR/pERK have been found to be increased in CD fibroblasts derived from both the skin and intestines of GFD–CD patients [39,42]. Structural alterations have been demonstrated in CD fibroblasts involving vesicular trafficking, focal adhesion [50] and cellular distribution of tTg [51].

Antigen-presenting cells have an important role in the pathogenesis of the disease. CD dendritic cells derived from CD patients have been studied to identify differences between patients and controls. Alterations of the cell shape in CD dendritic cells have been demonstrated to be dependent on Rho GTPase activity [52]. In an attempt to link genetic expression and function to a specific CD cellular phenotype, two genes found to be related through expression studies to CD, namely LPP (lipoma-preferred partner) and ARGHAP31, have been studied in CD fibroblasts and dendritic cells, respectively. LPP, a major regulator of focal adhesion function, has been found altered in CD fibroblasts [50] with increments in focal adhesion dynamics and phosphorylation in CD fibroblasts. Similarly, ARGHAP31 (Rho GTPase-activating protein 31), a Rho GTP regulator, has been found to be decreased in CD dendritic cells, together with increased Rho GAP activity and altered cell shape [52].

In conclusion, studies on cellular compartments far away from the intestines and intestinal organoids, where persistent inflammation and other functional and structural alterations have been demonstrated, point to the hypothesis that these lesions are “constitutive”. All together, these studies define a “celiac cellular phenotype” that renders the cells more prone to gliadin activity, which insists on the same pathways already altered in CD cells. This hypothesis is also sustained by studies on the natural history of the disease in at-risk populations, where markers of inflammation are present even before gluten introduction.

### 2.3. Exogenous Pro-Inflammatory Factors

In this paragraph, we briefly review the pro-inflammatory effects of gliadin, gliadin peptides, other wheat proteins, the Western diet and viral infections in CD patients (Table 2).

#### 2.3.1. Gliadin and Gliadin Peptides

Wheat is one of the most consumed cereals worldwide. Wheat flour contains in the endosperm grain storage proteins that are only partially digested by the human endopeptidases [54]. The gliadin P31-43 peptide resulted in being one the least digested gliadin peptides in vitro [54]. Moreover, it has been shown to induce several different biological effects in vitro and in vivo in intestinal epithelial cells and other cellular compartments. Most of the gliadin peptide P31-43 effects and structural peculiarity have been described elsewhere [54]. In vitro intestinal biopsies from CD patients both on a GFD and GCD show that this same peptide can induce structural alterations such as delay of endocytosis, induction of proliferation, alterations of innate immunity (IL15R-alpha, Interferon -alpha, INF-alpha) and inflammation markers (NFkβ) [54]. In DQ8 mice alone or, even better, in combination with viral ligands such as poli:IC (polyinosinic:polycytidylic acid), it can induce inflammation and alteration of the intestinal mucosa [54]. Moreover, gluten challenge in vivo on GFD–CD patients has an effect on the same pathways found altered before the challenge [25,26,40,54]. In vivo studies have demonstrated that the amount of ingested gluten/kilogram/day from 12 to 36 months was significantly greater in infants who developed celiac disease compared to those who did not up to 6 years of age, confirming that the amount of gluten ingested in the second year of life is a significant risk factor for developing celiac disease [22]. P31-43 has effects that are stronger, more prolonged and at lower concentrations in CD biopsies and in cells with respect to control. In particular, CD fibroblasts both from skin explant and from intestinal biopsies present increased sensitivity to P31-43 stimuli for the production of inflammatory markers [36]. More recently, an increased sensitivity to P31-43 and the viral ligand loxoribine has been shown in organoids from CD patients with respect to controls [17].

#### 2.3.2. Amylase/Trypsin Inhibitors (ATIs)

Cereal grains are attractive to pests and pathogens because they have high contents of storage reserves (starch and protein). They have therefore evolved to contain a range of proteins that inhibit the hydrolytic enzymes of these organisms, including ATIs able to inhibit α-amylases from several insects. Wheat inhibitors of α-amylase and trypsin have been studied for over 40 years, resulting in extensive literature [81]. Using mass spectrometry and 2D electrophoresis, several different fractions have been identified that all together account for 3.4–4.1 mg/g in whole meal flour of bread wheat. Wheat ATIs are well characterized as wheat allergens, particularly in Baker’s asthma but also upon ingestion of food. In addition, they have been studied widely over the past few years because of putative roles in other adverse reactions to wheat consumption, including celiac disease, and non-celiac wheat/gluten sensitivity. ATIs induce tissue inflammation in intestinal biopsies both for CD patients and control through the activation of TLR4. Although the inflammation induced in intestinal biopsies is not specific to CD, ATIs are a good example of pro-inflammatory components of food [82]. Other components are present in wheats, apart from ATIs and gliadins, that should be taken into account when considering the pro-inflammatory potential of cereals [82].

#### 2.3.3. Western Diet

The Western diet (WD) is a diet strictly linked to the Western lifestyle, characterized by high calorie consumption including fats, refined sugars and high protein content food. This diet and lifestyle have been strictly correlated to low-grade inflammation, which is the common ground for several different chronic diseases [8]. Nutrients can have inflammatory effects, through different mechanisms, on the intestines, and they can cooperate in the onset of CD [82]. The analysis of the dietary patterns of children genetically predisposed to CD, followed from birth to 6 years of age [22], suggests that not only may gluten be a risk factor, but other “pro-inflammatory” nutrients such as those of the “Western style diet” can have a role. The diet of the infants who developed CD in this cohort was significantly different from those who did not. The patients that develop active CD later on had a higher intake of carbohydrates, particularly oligosaccharides (sugars), and a lower intake of legumes, vegetables and fruit. In other words, those who became celiac patients followed, in the first few years of life, a diet more similar to the Western diet than to the Mediterranean diet (MD) [22].

#### 2.3.4. Viral Infections

The link between infections and CD has been established clinically by several large-scale, population-based cohort studies [61]. Some observations recently point to a cooperative mechanism between gliadin and infections that induce tissue inflammation. Gliadin itself can be an environmental trigger for CD both for its intrinsic characteristics and for the increased amount present in the diet [40]. However, a study on CD patients with genetic risk of the disease confirmed that enterovirus infections are associated with an increased risk of contracting the disease [61]. Moreover, cooperation between viral ligands of TLRs and gliadin has been shown in vivo in a mouse model and in vitro in intestinal epithelial cells, indicating cumulative effects between an aliment such as gliadin, which is frequently used, and viral infections [36,54]. All this may have important clinical implications that are discussed in the following section.

## 3. Future Prospective

What has been presented thus far connotes CD as a chronic inflammatory food disease, whose natural history, studied prospectively in subjects at risk, reveals an early inflammatory phase, even pre-gluten, which precedes the autoimmune response to gluten that is mediated by T cells. A CD cellular phenotype has also been identified, most likely genetically/epigenetically determined, which is characterized by constitutive cellular alterations that could generate a greater sensitivity of the celiac cells to some pro-inflammatory environmental factors (foods, gluten itself, infections, etc.) (Figure 1).

These new aspects of the natural history of CD are potentially valuable for future studies aiming at preventive strategies. The evaluation of the inflammatory profile of patients opens up new intervention scenarios—interventions that must be early and aimed at the prevention of tissue inflammation. For an extensive discussion on the topic, see the review by Auriccho R. and Troncone R. in the journal, *Frontiers in Immunology* [18].

Here, we discuss how these new findings can be translated into clinical practice, mainly in the prevention of the disease. In particular, we discuss the possible mechanisms in the prevention of the disease:(a)an anti-inflammatory diet, such as the MD;(b)intestinal viral infections that could interfere with immune tolerance to gluten;(c)reduction of the gluten load during intestinal infections.

### 3.1. The Preventive Role of an Anti-Inflammatory Diet, such as the MD

MD is the generic name of the traditional dietary patterns of the individuals living in the Mediterranean region characterized by a great abundance and diversity of non-starchy vegetables, minimally processed whole-grain cereals, legumes, nuts, seeds and few animal proteins in the diet. Many intervention trials using the MD have also shown beneficial effects in the treatment of several metabolic diseases, including chronic inflammatory diseases [83].

The exact mechanisms by which a MD exerts its beneficial effects in lowering the risk of developing cardiovascular disease, certain cancers, and other metabolic conditions is not known. Protection against oxidative stress and inflammation, inhibition of nutrient-sensing pathways by specific amino acid restriction, and gut microbiota-mediated production of metabolites influencing metabolic health are among the most important mechanisms that can mediate the pro-health effects of the MD.

A good adherence to the MD since the first months of life can prevent CD in the population at risk [22,82]. A (prudent) MD (high intake of vegetables, vegetable oils, pasta and grains, and low consumption of refined cereals and sweet beverages) at 1 year was associated with significantly lower CD autoimmunity by 6 years of age. The protective effect of the MD on the risk of developing CD may be explained by the anti-inflammatory potential of this diet, also partially mediated by an interplay with gut microbiota [82]. Taken together, these data suggest that an important clinical practice to prevent CD could be the introduction of the MD early in life, at weaning [84], as a general way to prevent inflammation and dysbiosis.

### 3.2. The Prevention of Intestinal Viral Infections That Could Interfere with Immune Tolerance to Gluten

Based on several large-scale, population-based cohort studies, a link between infections (viral and bacterial) and CD has been clinically established. The Teddy study [85] on 6327 children and the Norwegian Mother and Child Cohort study on 72,921 [86] children showed that early life infections may play a role in CD development and that *rotavirus* vaccination reduces the risk of CD [85,87]. Other studies also confirmed the role of infections and in particular *enterovirus* and *rotavirus* infections in the onset of CD in populations at risk [88,89]. Moreover, CD patients have a higher titer of antibodies against *reovirus* or human *adenovirus* serotype 2 [90,91]. The protective role of anti-*rotavirus* vaccination still needs to be completely established, but at least theoretically, the vaccine could prevent or delay the onset of the disease in children at risk [87].

### 3.3. The Reduction of Gluten Load during Intestinal Infections

The intestinal cells, particularly enterocytes and dendritic cells, can sense nutrients and respond by activating pro-inflammatory or anti-inflammatory mediators, partially through the same mechanisms that can recognize viruses or bacteria (i.e., TLRs) [61].

A study on CD patients at genetic risk of the disease has confirmed that *enterovirus* infections are associated with increased risk and has highlighted that further increased risk is conferred by interactions between *enterovirus* and higher gluten intake in the diet. By metagenomics of the fecal virome of at-risk CD patients, it was demonstrated that a cumulative effect of *enterovirus* and gluten amount can increase the risk of CD autoimmunity [92].

Moreover, a cooperation between viral ligands for TLRs and gliadin has also been shown in vivo and in vitro, suggesting that together with viral infections, food proteins are able to mimic and potentiate the innate immune response to viruses and can trigger an autoimmune disease such as CD [54,61]. In this context, it could be reasonable to reduce the inflammatory insults during *enterovirus* infections by introducing a gluten-free diet for preventive purposes in subjects at genetic risk for CD as a good clinical practice.

All these observations can be translated to other inflammatory diseases such as IBDs (inflammatory bowel diseases). In addition, for IBDs, a pro-inflammatory phase that precedes the onset of the disease has been described [93,94]. Dietary attitudes and Crohn’s disease [95] have a strong link. A study by Khalili et al. [96] confirmed that greater adherence to the MD was associated with a significantly lower risk of later-onset Crohn’s Disease. Moreover, it has been shown that there is an association between MD adherence and improved quality of life in a population of IBD patients. Furthermore, healthy lifestyle, including MD, has been shown to be most beneficial for elderly IBD patients. Overall, the MD is regarded to have a high potential to modulate gut inflammation and to be a therapeutic and preventive tool for IBD [96].

## 4. Material and Methods

We have selected the most recent literature on CD, at risk CD patients, GFD patient biopsies, inflammation markers, wheat proteins and diet in CD. Citing 79/96 references from the last 20 years and 50/96 from the last 5 years.

## 5. Conclusions

In celiac disease, inflammation plays a pivotal role, because there is an inflammatory pre-clinical set up that precedes the disease on which various pro-inflammatory agents from the environment can insist, including gluten. The clinical history of the disease seems to shift from pre-clinical inflammation to an autoimmune response and consequently to the onset of the disease. Historically, the focus of the research in CD has been the T cell response, perhaps neglecting other aspects that are recently coming to light, such as the state of inflammation that precedes the disease, which is also independent from the introduction of gluten.

From the point of view of preventing the disease, this pre-inflammatory state may be easier to modulate than the more complex adaptive response. Some clinical studies such as studies evaluating diet in children at risk already give indications that modulating this inflammatory state with Mediterranean-type diets has an effect on the onset of the disease. Similarly, preventing intestinal viral infections could play an important role in prevention. All this assumes a more general interest if we consider the other chronic inflammatory diseases such as IBDs and diabetes, where knowledge of the natural history of the disease in at-risk subjects followed from the early stage of life and early intervention on the state of inflammation of the subject could have an important impact on the health of the child and the adult.

## Figures and Tables

**Figure 1 ijms-23-07177-f001:**
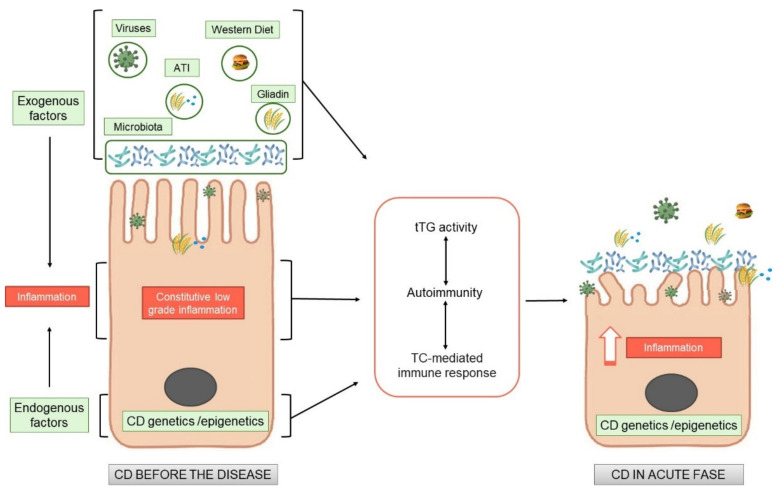
Both exogenous and endogenous factors can generate low-grade chronic inflammation in CD, initiating a series of events that will eventually induce an intestinal lesion.

**Table 1 ijms-23-07177-t001:** Endogenous, constitutive alterations of several pathways have been described in different cellular models of CD. Most of these endogenous alterations can predispose one to inflammation. Some of these constitutive alterations can be regarded as biomarkers of CD. CD: celiac disease; Wnt: Wingless and Int 1; NFkB: nuclear factor kappa-light-chain-enhancer of activated B cells; EGFR: epithelial growth factor receptor; ERK: extracellular signal-regulated kinases; ECM: extracellular matrix; pNFkB: phosphorylated (active) form of NFkB; pERK: phosphorylated (active) form of ERK; IL1 beta: interleukin beta 1; IL6: interleukin 6; LPP: lipoma-preferred partner; IL15: interleukin 15; IL15R alpha: IL15 receptor alpha; tTg: tissue transglutaminase.

Endogenous Factors	
Models Investigated	Pathways Described
**Children at risk of CD** **(before gluten introduction)**	(a) Lipid profiles in the blood [19,20] (b) Impaired growth of children [21] (c) Inflammation markers in the blood [22,23] (d) Gut microbiota changes [24]
**GFD–CD biopsies**	(a) Altered expression of genes related to: brush border assembly, development, transport cell cycle [25] (b) Wnt signaling alterations [25] (c) Epithelial inflammation, crypts hyperplasia, low grade inflammation in the serum [26,27] (d) NFkB pathway alterations [28,29,30,31] (e) Innate immunity activation [32,33,34,35,36,37,38] (f) EGFR/ERK mediated proliferation in the crypts [39,40,41,42] (g) Tight junctions and permeability alterations [6,43,44,45,46,47] (h) Impairment of vesicular trafficking [39]
**Intestinal organoids** **derived from CD patients**	(a) Increased staminality and permeability [48] (b) Increased inflammasome and innate immunity genes [48] (c) ECM genes decreased [49] (d) Increased pNFkB, pERK, IL beta, IL 6 [17]
**Nonintestinal cells** **from GFD–CD patients**	(a) In fibroblasts: Cell shape [39,42] Increased inflammation and innate immunity markers [39] Focal adhesions markers and LPP altered expression [50] Increased IL15 and IL15R alpha expression [39] Differential features of tTg [51] In dendritic cells: Cell shape alterations [52]

**Table 2 ijms-23-07177-t002:** Exogenous environmental factors can activate pro-inflammatory pathways in vivo and in vitro in CD cells. CD: celiac disease; Poli:IC: polyinosinic:polycytidylic acid.

Exogenous Pro-Inflammatory Factors	
Pro-Inflammatory Factors	Pathways Involved
**Gliadin and gliadin peptides**	(a) In mice: induce inflammation and alteration of the villi/crypts axis [54] (b) In vitro: delay of endocytosis, proliferation, alterations of innate immunity [54] (c) In vivo: the amount of gliadin ingested is higher in infants that develop CD [22]
**ATIs** **(Wheat Amylase Trypsin Inhibitors)**	(a) Induce inflammation in intestinal biopsies [81]
**Western diet**	(a) Mediterranean diet (MD) can prevent CD in at-risk population [22,82]
**Infections**	(a) In vivo: enteroviruses are associated with increased risk of CD [61] (b) In vitro: viral ligand loxoribine and gliadin cooperate to induce innate immunity activation [36] (c) In mice: cooperation between gliadin peptide P31-43 and Poli:IC to induce intestinal damage and innate immunity [54]

## Data Availability

Not applicable.

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
