# Peer review of "Pivotal Role of Inflammation in Celiac Disease"

_ijms, 2022, doi:10.3390/ijms23137177_

Round 1

Reviewer 1 Report

This manuscript discusses the pivotal role of inflammation plays in the celiac disease. The review of the topic is relevant to clinical practice because it opens a debate on new directions of study. I recommend the major revision of the manuscript, due mainly to the lack of depth of the presented figures that detract from the subject studied in the manuscript. The main arguments are listed below:

1) The title does not correlate with the cornerstone of the review since the study questions the importance of inflammation and does not presume that it is a chronic inflammatory disease and as a secondary fact the epithelium is damaged.

2) The authors should include gluten as keyword and even microbiota.   

3) Figure 1 does not provide to the reader additional information from the text of the manuscript. Please remove it.

4) The information contained in Figure 2 should be in a table since nothing is illustrated.

5) There is a lack of a figure showing some of the most striking results in a graphical form that even shows the possible relationship between them.

6) Do not forget to include in the figure captions each of the abbreviations used.

7) What do the authors mean by the acronym GFD-CC? should it read CD?.

8) There is a lack of uniformity in the text in relation to celiac disease and coeliac disease (page 11, line 471) and use of acronym cross the text (example CD, GFD).

9) ATIs acronyms should be defined (page 11, line 461).

10) The use of Rotavirus and Enterovirus when referring to the genus should be in italics (page 12, line 558).

11) Perhaps it would be interesting to comment why the effect of Adenovirus infections has not been correlated with an increase in the number of celiac debutants in relation to the inflammatory process.

12) The acronym Mediterranean diet is not well defined (page 11, line 492).

13) Moderate English changes are required.

Reviewer 2 Report

This is an interesting review aiming to discuss the recent literature in celiac disease (CD), analyzing the main role of the inflammation in the natural history of the disease, supported by a cellular fragility that renders the cells more sensitive to gluten and other pro-inflammatory agents. The authors aimed at discussing: (1) the available evidence of a constitutive fragility of the intestine in CD; (2) the susceptibility of coeliac patients and cells to gliadin and other pro-inflammatory triggers; and (3) future prospective.

The manuscript is well written and organized. However, some important topics should be recalled.

-in my opinion, the authors should recall and discuss the impact of altered intestinal permeability in the secondary autoimmune phenomena as well as the extraintestinal manifestations of celiac disease. This is a topic of major interest in a review addressing the link between inflammation, mucosal damage, and clinical features. In particular, CD patients may exhibit autoantibodies directed against extraintestinal antigens such as antineuronal and antiganglioside antibodies (more prevalent in patients with neurological disorders) as previously reported (Sera of patients with celiac disease and neurologic disorders evoke a mitochondrial-dependent apoptosis in vitro. Gastroenterology. 2007 Jul;133(1):195-206; Anti-ganglioside antibodies in coeliac disease with neurological disorders. Dig Liver Dis. 2006 Mar;38(3):183-7.).

-it has been also previously demonstrated that CD patients on a gluten-containing diet and severe mucosal damage (villous atrophy) frequently present anti-actin IgA antibodies that display high specificity for celiac disease, and disappear after gluten-free diet, as previously reported (Anti-actin IgA antibodies in severe coeliac disease. Clin Exp Immunol. 2004 Aug;137(2):386-92.).

-Finally, a link between atopy and CD has been suggested by an Italian study (Prevalence of silent coeliac disease in atopics. Dig Liver Dis. 2000 Dec;32(9):775-9).

Round 2

Reviewer 1 Report

Dear authors, 

I appreciate the effort made to improve the article. However, there are still some things that have not been adequately understood:

The concept of a table is by definition an arrangement of data in rows and columns. I consider that tables 1 and 2 should be modified and manually enter the information in the rows. In addition, in the case of tables, the table legend is placed at the top of the table.

Page 5, line 190: A space is required between nucleotide and polymorfisms and another after the parenthesis

Page 7, line 294-295:  A space is required after the parenthesis

Page 7, line 301: Myxovirus should be in italics

Page 8, lines 340 and 345: celiac disease should be abbreviated as CD

Page 8, line 346: gluten-free diet should be abbreviated asd GFD

Page 8, lines 361-362: give space after the parentheses

Page 10, lines 433: give space after the parentheses

Page 11, line 483: give space after the parentheses

Page 13, line 609: The abbreviation has already been previously entered in page 12, line 515. 

Reviewer 2 Report

The manuscript is now improved and very well discussed. It can be now accepted for publication.

Author Response

Attachment
